# Leaf Rust Resistance Genes in Wheat Cultivars Registered in Russia and Their Influence on Adaptation Processes in Pathogen Populations

Elena Gultyaeva *, Ekaterina Shaydayuk and Philipp Gannibal

All Russian Institute of Plant Protection, 196608 St. Petersburg, Russia; eshaydayuk@bk.ru (E.S.);
phbgannibal@yandex.ru (P.G.)
* Correspondence: egultyaeva@vizr.spb.ru

**Abstract:** The main growing regions for winter wheat in the Russian Federation are the North Caucasian, Central Black Earth, and Central agroecological regions. Spring wheat crops dominate in the Urals, Volga region, and Western Siberia. Wheat leaf rust, caused by *Puccinia triticina*, is an important disease, impacting greatly on wheat production. In Russia, the disease was an annual problem until 2010 but has since been more effectively controlled. However, changes in virulence in pathogen populations may arise from climate change, evolving cropping practices, intense use of chemical protectants, and an increase in the release of resistant cultivars. In the 2000s, the State Register of the Russian Federation included an increase in the number of winter and spring wheat cultivars resistant to leaf rust. However, successful genetic protection requires a diversity of cultivars with different resistance genes (*Lr* genes). Studies by the All Russian Institute of Plant Protection identified *Lr* genes in Russian cultivars' phenotypes and molecular markers. In addition, the prevalence of virulence in pathogen populations was studied and the influence of the cultivar used in wheat production on the changes in these populations was evaluated. This paper reviews research on the genetic diversity of winter and spring wheat cultivars included in the State Register of Russia from 2000 to 2020 and analyzes their impact on the prevalence of virulence in pathogen populations. These data demonstrate the continuous evolution of *P. triticina* in response to wheat breeding efforts. Populations of the pathogen showed higher variability in regions where pathotype-specific resistance cultivars were commonly grown.

**Keywords:** *Lr* genes; *Puccinia triticina*; resistance; *Triticum aestivum*

## 1. Introduction

Wheat leaf rust, caused by *Puccinia triticina*, is an important disease that affects production in many agroecological regions of the Russian Federation. Prior to the 2000s, two to three times per decade, damaging epidemics would occur in the North Caucasian region, resulting in yield losses of 30–35%. Similar yield losses from leaf rust were even more frequent in the Central and Volga regions, occurring five to seven times per decade. In the Central Black Earth region, and the Volga-Vyatka and Ural regions, losses were lower, at 15–20%, with damaging epidemics four and five times per decade, respectively [1]. From 2010, the disease began to lose its significance as, in many regions of Russia (Volga and Western Siberia regions), stem rust began to predominate, and in the North Caucasus, it was yellow rust [2,3]. A decrease in the importance of *P. triticina* in 2000 was also noted in other countries, including those neighboring Russia (Kazakhstan, Armenia, Azerbaijan, Georgia, Kazakhstan, Kyrgyzstan, Turkmenistan, Uzbekistan, etc.) [4,5]. These shifts in pathogen and disease prevalence created new challenges, especially as regards climate change, evolving cropping practices, intense use of chemical protectants, and an increase in the release of resistant cultivars.

Over the last decade, there has been great progress in the breeding of bread wheat in Russia. The number of winter wheat cultivars included in the State Register of Breeding Achievements and recommended for cultivation in Russia in 2020 increased fourfold compared to the mid-1990s (333 in 2020 vs. 82 in 1996) and spring wheat cultivars increased by two and half times (261 vs.112) [6]. The development of strong resistance in wheat to leaf rust and a regional scale requires genetic diversity of cultivars in the resistance genes deployed. Vertical resistance (oligogenic) conferred by single dominant resistance genes can be readily overcome by changes in pathogen virulence. The main benefit of oligogenic resistance is to suppress the primary infection of the pathogen. In contrast, horizontal (polygenic) resistance is nonspecific and acts against all pathotypes of the pathogen. Horizontal resistance is considered to persist longer than vertical resistance, as the pathogen population is less likely to accumulate sufficient virulence mutations to overcome this polygenic resistance [7]. The likelihood of new pathotypes overcoming oligogenic resistance of cultivars is greater when large areas are sown with cultivars with the same or similar resistance genes.

The main production regions for winter wheat in Russia are the North Caucasian, Central Black Earth, and Central regions, and, for spring wheat, it is the Volga (Low, Middle, and Volgo-Vyatca), Ural, and Western Siberia regions (Figure 1). Russia's large size means that there is a wide diversity of agricultural environments. Consequently, regional breeding institutions have developed a diversity of cultivars that mirror the diversity of the growing conditions. Over the last decade in Russia, breeding has been enhanced by the adoption of marker-assisted breeding, and this innovation has delivered increased efficiency for the development of new cultivars. The use of molecular markers has enabled breeders to select superior wheat genotypes with rust resistance that would have been difficult to achieve based on phenotyping alone, given the responsiveness of pathogen populations to the resistance genes deployed in the widely grown cultivars. In particular, it has also facilitated the combination of genes into desirable agronomic backgrounds to establish synergies between resistance genes, even including those that are no longer effective on their own [8].

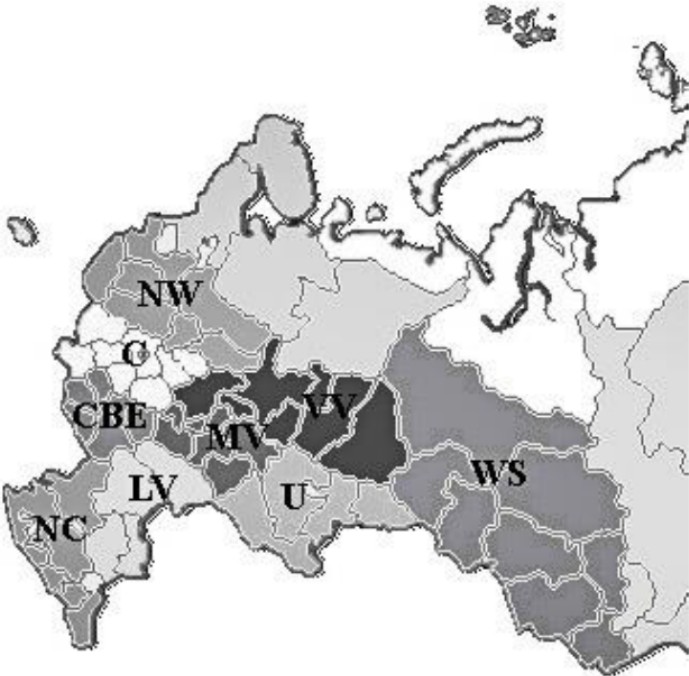

**Figure 1.** Russian agroecological regions. NW, Northwestern; NC, North Caucasian; CBE, Central Black Earth; C, Central; LV, Low Volga; U, Ural; WS, Western Siberian; MV, Middle Volga; and VV, Volgo-Vyatka regions.

Managing the genetic resistance of wheat is impossible without pathogen population studies. In Russia, such studies have been ongoing since the 1930s. The main changes in the virulence of *P. triticina* in Russia, as in other countries, are driven by the production of genetically similar wheat cultivars over large areas. A well-known example of this is the epidemic of *P. triticina* in 1973 in the widely grown, previously resistant cultivars, Aurora and Kavkaz (with resistance gene *Lr26*), Skorospelka (with *Lr3*), and Bezostaya 1 (with *Lr34*), in the North Caucasus [9,10]. Intensification of grain production in the late 1980s resulted in a rapid loss of resistance in other previously resistant cultivars. For example, the cv. Yuna, widely grown in the North Caucasus, became susceptible within 5 years of release [10]. The development of virulence in the pathogen to resistant cultivars protected by previously effective *Lr* genes also occurred in regions of spring wheat production in Russia. In the 1970s to 1980s, *Lr9* (from *Aegilops umbellulata*), *Lr19* and *Lr24* (from *Thinopyrum elongatum*), and *Lr23* (from *Triticum durum*) were the most effective genes against wheat leaf rust and were widely used in the breeding of spring wheat. In addition, amphidiploids were obtained from various sources including *Aegilops speltoides*, *Aegilops tauschii*, *Thinopyrum intermedium*, and *Triticum timopheevii* [10–13]. *Lr19* and *Lr23* were widely used in breeding programs in the Volga region, with the first cultivars with these genes released in the 1980s. However, by the mid-1990s, these genes were no longer effective due to changes in pathogen virulence [14,15]. *Lr9* was widely used in breeding in Western Siberia and the Urals, but in 2007, the first *Lr9*-virulent isolates were identified [16]. Since 2000, there has been further progress in Russia in the development of cultivars resistant to leaf rust [10,17,18]. This progress can be seen in the findings of studies conducted in 1996 by the All Russian Institute of Plant Protection on seedling resistance in 61 cultivars of winter wheat and 100 cultivars of spring wheat selected from the State Register. Among these, only two resistant winter wheat cultivars (Dakha and Polovchanka) and five spring wheat cultivars (L503, L505, Prokhorovka, Samsar, and Tertsiya) were identified, with the proportion of leaf rust-resistant cultivars in the Register at less than 4%. However, a similar analysis in 2005 revealed an increase in spring wheat cultivars with vertical resistance (up to 15%). These included spring wheat cultivars Belyanka, Ekada 6, Fora, Dobrynya, Obskaya 14, Piramida, Sonata, Tulaykovskaya 5, Tulaykovskaya 10, Tuleyevskaya, Yuliya, Volgoural'skaya, and Yershovskaya 32, and the winter wheat cultivar Splav. Analysis of 120 winter and 60 spring bread wheat cultivars included in the State Register in 2006–2011 revealed vertical resistance to leaf rust in 3% of winter wheat cultivars (Ayvina, Bogdanka, Nemchinovskaya 24, and Poema) and 25% of spring wheat cultivars (Altayskaya 110, Altayskaya 530, Chelyaba stepnaya, Chelyaba yubileynaya, Favorit, Kinel'skaya Niva, Kinel'skaya otrada, Lebedushka, Mariya 1, Novosibirskaya 44, Omskaya 37, Sibakovskaya yubileynaya, Tulaykovskaya zolotistaya, Udacha, Tulaykovskaya 100, and Voyevoda) [18,19]. The increase in the number of cultivars resistant to leaf rust continued from 2010 to 2020.

The All Russian Institute of Plant Protection has studied *Triticum aestivum–P. triticina* interactions since the 1930s [20]; mainly, these are studies on host resistance in wheat breeding lines and cultivars as seedlings in controlled environments and adult plants in the field. Studies on the pathogen included analysis of virulence and pathotype composition in wheat-growing areas of Russia. Up until the 1980s, the study of populations was done using Mains and Jackson's differential cultivars. In the 1980s, the Thatcher isogenic lines with *Lr* genes (*TcLr*) were also used. In the 1990s, Tyryshkin and Mikhailova [21] developed a new set of wheat differentials that helped to demonstrate the existence of different regional populations of *P. triticina* in Russia (European, Asian, and Caucasian). In the 2010s, this finding was confirmed using molecular markers [22].

Since 2005, phytopathological methods for studying host–pathogen interactions have been supplemented with molecular methods [17,18,22]. Currently, molecular markers are used by the Institute to identify highly effective genes (*Lr24*, *Lr25*, *Lr28*, *Lr29*, *Lr41*, *Lr50*, *Lr47*, *Lr51*, and *Lr66*), partially effective genes (*Lr9* and *Lr19*), adult plant resistance genes (*Lr21*, *Lr34*, *Lr35*, and *Lr37*), and the now ineffective genes (*Lr1*, *Lr3*, *Lr10*, *Lr20*,

and *Lr26*) (Table 1). The information obtained allows the determination of the distribution of *Lr* genes across cultivars grown in Russia, characterization of the genetic diversity of the modern wheat lines, and assessment of their potential influence on the adaptation processes of pathogen populations.

**Table 1.** Markers used for identification wheat rust resistance genes (*Lr* genes).

| *Lr* Gene | Markers | Marker Type | Size of Amplified Marker Fragments, bp | Reference |
|---|---|---|---|---|
| *Lr1* | WR003 F/R | PCR | 760 | Qiu et al. [23] |
| *Lr3a* | Xmwg798 | STS | 365 | Herrera-Foessel et al. [24] |
| *Lr9* | J13 | STS | 1100 | Schachermayr et al. [25] |
| | SCS5 | SCAR | 550 | Gupta et al. [26] |
| *Lr10* | F1.2245/Lr10-6/r2 | STS | 310 | Chelkowski et al. [27] |
| | Lrk10-6 Lrk10-D | STS | 282 | Schachermayr et al. [28] |
| *Lr19* | Gb | STS | 130 | Prins et al. [29] |
| | SCS265 | SCAR | 512 | Gupta et al. [30] |
| *Lr20* | STS638 | STS | 540 | Neu et al. [31] |
| *Lr21* | Lr21F/R | STS | 669 | Fritz [32] |
| *Lr24* | Sr24 ≠ 12 | STS | 500 | Mago et al. [33] |
| | Sr24 ≠ 50 | STS | 200 | |
| | SCS73 | SCAR | 719 | Cherukuri et al. [34]; Prabhu et al. [35] |
| | J09 | STS | 310 | Schachermayr et al. [36] |
| | SCS1302 | SCAR | 607 | Gupta et al. [37] |
| | S1326 | SCAR | 613 | |
| | SCOAB-1 | SCAR | 365 | |
| *Lr25* | Lr25F20/R19 | SCAR | 1800 | Procunier et al. [38] |
| *Lr26* | SCM9 | PCR | 207(1BL.1RS) | Weng et al. [39] |
| | | | 228 (1Al.1RS) | |
| | iag 95 | STS | 1000 | Mago et al. [40,41] |
| *Lr28* | SCS421$_{570}$ | SCAR | 570 | Cherukuri et al. [42] |
| *Lr29* | Lr29F24 | SCAR | 900 | Procunier et al. [38] |
| *Lr34* | csLV34 | STS | 150 | Lagudah et al. [43] |
| | L34DINT9F/L34PLUS | PCR | 517 | Lagudah et al. [44] |
| *Lr35* | Sr39 F2/R3 | SCAR | 900 | Gold et al. [45] |
| | BCD260F1/35R2 | STS | 450 | Seyfarth et al. [46] |
| | Sr39#22r | STS | 800 | Mago et al. [47] |
| *Lr37* | Ventriup/LN2 | STS | 285 | Helguera et al. [48] |
| *Lr41(39)* | GDM35 | SSR | 190 | Pestsova et al. [49]; Brown-Guedira, Singh [50] |
| *Lr47* | PS10 | PCR | 282 | Helguera et al. [51] |
| *Lr50* | Xgwm382 | SSR | 139 | Brown-Guedira, Singh [52] |
| | Xgdm87 | SSR | 110 | |
| *Lr51* | S30-13L/AGA7-759 | CAPS | 422 + 397 | Helguera et al. [53] |
| *Lr66* | S13-R16 | SCAR | 695 | Marais et al. [54] |

The objective of this paper is to review these studies and provide an overview of the current status of wheat leaf rust resistance in Russia.

## 2. Genetic Diversity of Winter and Spring Bread Wheat Cultivars in Russian Agroecological Regions and the Virulence of *Puccinia triticina*

**North Caucasus region**. The North Caucasus region, located in the southern part of European Russia, is the main grain-producing region. The State Register of Breeding Achievements recommends over 50% of listed cultivars as suitable for growing in the North Caucasus region. Before the 2000s, most of the North Caucasian cultivars of wheat were characterized as being susceptible to leaf rust. This was a result of the large areas sown with single cultivars and the intensity of wheat production. In the 1990s, a policy for the development and deployment of cultivars with different types of genetic resistance (both major genes and adult plant resistance genes) was adopted. This approach prevents a single cultivar from being dominant, as in the past, as it prohibits any single cultivar from being sown over more than 15–20% of the wheat area. This strategy is considered to be the main reason for the gradual reduction in leaf rust occurrence in the region in the 2000s [10].

The current Register suggests that more than half of the winter wheat cultivars grown in the North Caucasus have varying types of field resistance to leaf rust [6]. Using molecular markers, low and partially effective *Lr* genes in these cultivars were identified, except for the cv. Gerda, which had the effective juvenile gene, *Lr9*. Since 2015, cultivars with the adult plant resistance gene, *Lr37* (Markiz, Morozko, Svarog, in combination with *Lr1*; Gomer with *Lr1*; and Graf with *Lr1* and *Lr10*), were recommended for cultivation in the region for the first time. This gene effectively protects wheat from leaf rust in the North Caucasus.

In other resistant cultivars, there was a wide distribution of the now ineffective genes, *Lr1*, *Lr3*, *Lr10*, and *Lr26*, and the partial resistance gene, *Lr34* (Table 2). These genes were found either alone or in varying combinations in the cultivars. Individually, they lost their effectiveness, but in combination, they provide an increase in the level of field resistance. This is consistent with the data of Dakouri et al. [55], who showed that wheat genotypes with three or more of these ineffective genes (*Lr1*, *Lr3*, *Lr10*, and *Lr20*) had higher resistance in the field than those with only one or two of these genes. The presence of the partial resistance gene, *Lr34*, additionally strengthens the effect of these juvenile genes.

In 2000–2020, the structure of the North Caucasian populations of *P. triticina* did not undergo any significant shifts in terms of virulence, which was probably due to balanced genetic protection and the high diversity of cultivars grown in the region. The number of virulence alleles in the North Caucasian populations of *P. triticina* was lower than in other Russian populations. *Lr9*, *Lr19*, *Lr24*, *Lr28*, *Lr29*, *Lr41*(=39), *Lr42*, *Lr45*, *Lr47*, *Lr50*, *Lr51*, *Lr53*, and *Lr57* were highly efficient in the region. These genes (except for *Lr9* and *Lr19*) were also effective in other grain-producing regions of Russia. *Lr1* showed an increase in virulence by 2010 and had higher value in subsequent years [17]. Annual variation in virulence prevalence without clear dynamics was observed in genotypes with *Lr2a*, *Lr2b*, *Lr2c*, *Lr15*, *Lr20*, and *Lr26*. The number of virulence alleles in the North Caucasian populations of *P. triticina* was lower than in other regional populations in Russia. Genes such as *Lr26*, *Lr34*, and *Lr37*, widespread in North Caucasian cultivars, are linked to the effective stem rust resistance genes, *Sr31*, *Sr57*, and *Sr38*, which has provided stable genetic protection of wheat against stem rust in this region.

**Central European regions.** The Central European regions include the Central Black Earth, Central, and Northwestern regions. In the Central Black Earth region, winter wheat cultivars predominate; both winter and spring wheat cultivars are grown in the Central and Northwestern regions. Approximately 25% of winter wheat cultivars and 18% of spring wheat cultivars in the State Register are recommended for these regions [6]. Most of these cultivars are susceptible to leaf rust. Over the years, winter wheat cultivars such as Inna, Mironovskaya 808 (*Lr3a*), Mironovskaya yubileynaya, Moskovskaya 39 (*Lr1*), Moskovskaya 70 (*Lr1*), Pamyati Fedina, Galina, TAU (*Lr1*), Bezenchukskaya 3, Belgorodskaya 12, Belgorodskaya 16, Skipetr (*Lr10*, *Lr26*),

Zarya, and spring Moskovskaya 35 (*Lr1*), Irgina, Iren, Ester (*Lr10*), Trizo, (*Lr20*), Dar'ya (*Lr20*), Leningradskaya 97, Mis (*Lr10*), Saratovskaya 29 (*Lr10*), and Lada (*Lr10*) were widely grown in this European part of Russia.

**Table 2.** Resistance genes *Lr34* and *Lr26* in the North Caucasian winter bread wheat cultivars recommended prior to 2000, from 2001 to 2010, and from 2011 to 2020.

| Recommendation Period * | *Lr34* | *Lr26* | *Lr34* and *Lr26* |
|---|---|---|---|
| Before 2000 | Bezostaya 1, Podarok Donu, Tarasovskaya ostistaya; in combination with *Lr3:* Don 93, Donskaya bezostaya, Donskaya yubileynaya, Donskoy mayak, Zernogradka 9 | | in combination with *Lr10*: Vita |
| 2001–2010 | Bulgun, Garant, Dominanta, Dzhangal', Konkurent, Kuma, Liga 1, Stanichnaya, Odesskaya 100, Pamyati Kalinenko, Rostovchanka 3, Yubileynaya 100, Zustrich, Severodonskaya;in combination with *Lr1*: Don 107, Gubernator Dona, Pisanka, Rapsodiya, Tristan, Yunona, Pervitza, Pisanka, Zimtra; in combination with *Lr3*: Arpha, Don 105, Deviz, Donskoy proctor, Prestizh, Skiphyanka, Tarasovskaya ostistaya, Zernogradka 10, Zernogradka 10, Viza;in combination with *Lr10*: Moskvich, Resurs Rostovchanka 5, Skpabitsa, Zimtra, Yumpa; in combination with *Lr1* and *Lr10*: Zimtra | Berezit, Bulgun, Fantasiya, Fortuna, Irishka, Krasota, Knyazhna, Maphe, Legenda, Selyanka, Umanka, Veda, Vostorg, Yashkulyanka, Polovchanka; a; in combination with *Lr1*: Pervitsa; in combination with *Lr10*: Bulgun, Gratziya, Kollega, Kupava | Veda, Doka, Tanya, Yesaul; Dzhangal', Gordyanka; in combination with *Lr10*: Ayvina, Sintetik, Vita; in combination with *Lr1*: Afina, Kseniya |
| 2011–2020 | Arsenal, Bezmezhna, Blagodarka odes'ka, Borviy, Boyarynya, Dmitriy, Donskaya Lira, Firuza 40, Kalym Kaprizulya, Maykopchanka, Missiya, Nakhodka, Niva Stavropol'ya Goduval'nytsya Odes'ka, Idilliya, Ovidiy, Poshana, Sekletiya, Sluzhnytsya odes'ka, Titona, Donera, Viktoriya 11, Vol'nuy Don, Vol'nitza, Zhavoronok, in combination with *Lr3*: Aksin'ya, Anastasiya, Asket, Izyuminka, Lidiya, Lilit; in combination with *Lr1*: Akapella, Driada1, Misiya odes'ka, Otaman, Proton, Vdala, Zagrava odesskaya, Knoppa, Vol'nitsa, Vol'nyy Don; in combination with *Lr1* and *Lr3*:V'yuga; in combination with *Lr1*, 1AL.1RS: Kokhana, Knyaginya Ol'ga; in combination with *Lr1*, *Lr10*: Anka, Bagira, Kuyal'nik, Lastivka odes'ka, Zhayvir, Zmina;in combination with *Lr3* and *Lr10*: Shef | Alekseich, Anastasiya, Armada, Chornyava, Donstar, Karavan, Kuren', Laureat, Ol'khon, Step', Timiryazevka 150, Vassa, Velena, Videya; in combination with *Lr1*: Akhmat, Gurt, Uryup, Vershina, Yuka; in combination with *Lr1 and Lr10*: Antonina, Bagrat, Kurs; in combination with *Lr3*: Iridasin in combination with *Lr10*: Veha, | Bezostaya 100, Podolyanka; Utrish, Zhiva; in combination with *Lr10*: Adel'; Paritet, in combination with *Lr1*: Duplet, Korona, Vid; in combination with *Lr3*: Vanya. |

* According to the State Register of Breeding Achievements of the Russian Federation [6].

The first winter wheat cultivar, Splav, with effective pathotype-specific resistance to leaf rust, was included in the Register in 2002 and recommended for cultivation in the Northwestern region. Molecular markers revealed the presence of *Lr9* and *Lr34* genes in Splav. In 2006 and 2013, two cultivars with *Lr9*, Nemchinovskaya 24 and Nemchinovskaya 9, were included in the Register and widely grown in the Central region. In 2010, two new resistant winter wheat cultivars, Bogdanka and Poema, were approved. The resistance of Bogdanka was due to the rye translocation 1AL.1RS and partial resistance gene *Lr34*. The genetic basis of the resistance of Poema is unknown. Molecular marker analysis has not revealed the presence of any known effective *Lr* genes from the Catalogue of Gene Symbols for Wheat [56].

In mid-2010, foreign spring wheat cultivars with the highly effective *Lr24* gene were recommended for the region, viz. Kanyuk (combined with 1AL.1RS and *Lr20*)

(SECOBRA Recherches SAS), KWS Akvilon, KWS Sunset (KWS LOCHOW Gmbh), and others. In 2012, the new spring wheat cultivar Kurier, moderately resistant to leaf rust, was recommended. Its resistance is provided by a combination of the now individually ineffective genes, *Lr1*, *Lr10*, and *Lr26*. Moreover, spring wheat cultivars with the *Lr19* gene (cvs L503, L505, and others) are recommended for the region.

The populations of *P. triticina* in the Central European regions are characterized by higher variability in virulence compared to those in the North Caucasian region. The widespread production of cultivars with *Lr9* in the region facilitated the emergence of virulent isolates of *P. triticina*. These were recorded for the first time in 2013. However, there was no significant increase in their prevalence in subsequent years. Isolates virulent to Tc*Lr19* had a higher prevalence in 2000–2008. In subsequent years, they were mainly detected in cultivars protected by this gene. Virulence to *Lr24* was also rare and was recorded only sporadically in 2007–2008 and 2013. The prevalence of virulence to *Lr2a*, *Lr2b*, *Lr2c*, and *Lr15* was higher before 2010 but has decreased slightly since. The widespread production of cultivars with *Lr1* in the region facilitated a gradual increase in the prevalence of virulence in the pathogen populations (59% in 2001–2005, 69% in 2006–2010, 93% in 2011–2015, and 100% in 2016–2020).

**Volga regions** The Volga regions (Lower Volga, Middle Volga, Volga-Vyatka) are a spring wheat production area because winter wheat is frequently killed during severe winters. The amount of precipitation is variable and large areas of crops in the region (especially the middle and lower Volga regions) are subjected to moisture stress during the growing season. Unlike the winter wheat breeding programs in Southern Russia relying primarily on slow-rusting, the strategy of rust resistance breeding in the Volga region is based on continuous screening for possible sources of resistance and incorporation of resistance genes from wild relatives and related wheat species [10]. The compositions of the cultivars grown in the three Volga regions differ significantly from each other. In the Volga-Vyatka region, the range of cultivars is similar to those grown in the Central European regions and the Urals. Most of these cultivars are susceptible to leaf rust. In the 2000s, cultivars such as Moskovskaya 39 (*Lr1*), Amir (*Lr10*), Ester (*Lr10*), Irgina, Iren, Krasnoufimskaya 100 (*Lr34*), Lada (*Lr10*), Prokhorovka (*Lr3*, *Lr26*, and *Lr10*), Priokskaya (*Lr1* and *Lr10*), Zarya, Ekada 70 (*Lr10*), L503 (*Lr19* and *Lr10*), and others were widely grown in the region.

Roughly 10% of cultivars in the State Register are recommended for the Lower Volga region. Cultivars with *Lr10*, *Lr26*, and *Lr19* (Al'bidum 28, Al'bidum 32, Pamyati Aziyeva, Saratovskaya 29, Saratovskaya 55, Saratovskaya 68, Saratovskaya 73, and Saratovskaya 74, with *Lr10*; Yugo-Vostochnaya 2, Yugo-Vostochnaya 4 and Prokhorovka combined with *Lr26* and *Lr10*; Aleksandrit and Dobrunya with *Lr19*; and L503 and L505 with *Lr19* and *Lr10*:) are widely grown in the region. In 1999, cv. Belyanka, with the highly effective resistant gene, *Lr6Agi1* from *Th. intermedium*, was included in the Register. This gene differs from the known genes included in the Catalogue of Gene Symbols for Wheat [56]. In 2007–2009, three more cultivars with this gene, Favorit, Voyevoda, and Lebedushka (the latter combined with *Lr19*) were approved for production in the Lower Volga region. *Lr6Agi1* is currently still effective in Russia, and no virulent isolates of *P. triticina* have been identified.

Approximately 15% of the cultivars in the State Register are recommended for the Middle Volga, including many cultivars with the *Lr19* gene, Ekada 113, Kinel'skaya Niva, Kinel'skaya 61 (combined with *Lr10*), Kinel'skaya yubileynaya, Samsar, Tulaykovskaya 110 (with *Lr3*), Tulaykovskaya 108, and Ul'yanovskaya 105 Volgoural'skaya, and Yuliya. In 2001, cv. Tulaykovskaya 5 with *Lr6Agi2*, which is highly resistant to leaf rust, was also approved for this region. *Lr6Agi2*, like *Lr6Agi1*, was transferred from *Th. intermedium*. Cultivars with *Lr6Agi1* and *Lr6Agi2* have a substitution in wheat chromosome 6D from chromosomes 6Agi and 6Agi2, which belong to the J (=E) subgenome of *Th. intermedium* [57]. In 2003–2007, the State Register included three more cultivars with *Lr6Agi2*, Tulaykovskaya 10, Tulaykovskaya zolotistaya (combined with *Lr3*), and Tulaykovskaya 100. These cultivars are currently commonly used as donors of leaf rust resistance in breeding programs

in the Volga region, the Urals, and Western Siberia. Virulence to *Lr6Agi2* has not been recorded in Russia. Additionally, cultivars with *Lr9* (Kinel'skaya otrada and Kinel'skaya 2010 combined with *Lr10*) are recommended for growing in the Middle Volga region.

The production of genetically protected cultivars in the Volga region drives the dynamics of its pathogen populations. Analysis of virulence consistently revealed virulence to *TcLr19* in all years investigated. The prevalence of virulence to *TcLr19* was higher before 2010 but subsequently decreased. *Lr9*-virulent isolates of *P. triticina* in the Middle Volga were recorded for the first time in 2013. However, no subsequent increase in their prevalence has been observed. Virulence to *Lr24* is rare but was recorded in 2008 in the Volga-Vyatka and Middle Volga regions. Virulence to *Lr1*, *Lr2a*, *Lr2b*, *Lr2c*, and *Lr15* in the Volga was higher than in other European populations of the pathogen but lower than in Asian populations [58].

**Ural region.** The Ural region is climatically subdivided into the Cis-Urals and Trans-Urals, with the latter being more arid. The watershed along the peaks of the Ural Mountains is the divide between Europe and Asia. Approximately 15% of the cultivars in the State Register are recommended for the Urals [6], but the recommendations differ for the Cis-Urals and Trans-Urals regions, in line with the composition of their pathogen populations. The virulence of *P. triticina* in the Cis-Urals (Bashkortostan) is similar to its populations in the Volga and Central European regions, whereas in the Trans-Urals (Chelyabinsk and Kurgan), the virulence is closer to that of Western Siberian.

The now ineffective genes (*Lr9*, *Lr26*, *Lr10*, and *Lr1*) are common in the spring wheat cultivars of the Urals. The first cultivars with the *Lr9* gene, Kvinta and Duet, in the South Urals, were released in the early 2000s. These cultivars are now widely used in breeding programs. New cultivars with the *Lr9* gene have been developed (Duet, Chelyaba 2, Chelyaba yubileynaya, Chelyaba rannyaya, Pamyati Ryuba and Chelyaba stepnaya combined with *Lr10*; Zauralochka with *Lr1*; Ariya with *Lr3*; and Iset 45 with *Lr3*). The widespread use of cultivars with *Lr9* in the region led to it becoming completely ineffective in 2010. Currently, any cultivars with only the *Lr9* gene are severely affected by leaf rust. The exception is Silach, which contains an effective combination of *Lr9*, *Lr26*, and *Lr10*. Silach has high resistance to leaf rust, both in seedlings and mature plants, because no populations of the pathogen in the region have virulence to both *Lr9* and *Lr26*, or both *Lr19* and *Lr26*. However, virulence to *Lr9* (or *Lr19*), and *Lr1*, *Lr2a*, *Lr2b*, *Lr2c*, *Lr3a*, *Lr3bg*, *Lr3ka*, *Lr10*, *Lr14a*, *Lr14b*, *Lr15*, *Lr17*, *Lr18*, *Lr20*, and *Lr30*, is widespread.

In 2012, the new, highly resistant wheat cultivar Chelyaba 75, with the gene *LrSp* transferred from *Aegilops speltoides* (also with *Lr1* and *Lr10*), was recommended for production in the South Urals [59]. *LrSp* is localized in the 2DS.2SL translocation [60] and differs from other known genes transferred from *Ae. speltoides* (*Lr28*, *Lr35*, *Lr36*, *Lr47*, *Lr51*, and *Lr66*). No virulence to Chelyaba 75 has been found in the Russian populations of *P. triticina*, despite its now extended period of production and expanding production area. Chelyabinsk Research Institute of Agriculture has distributed other cultivars with this gene, namely Chelyaba 80, Pamyati Odintsovoy, Odintsovskaya, and Eritrospermum 25908, for wider testing.

Cultivars with the *Lr26* gene are widely grown in this region. Over 30% of the cultivars recommended for the Ural region have this gene, including Alabuga, Boyevchanka (combined with *Lr10*), Gerakl, Grenada, Ingala, Izumrudnya (with *Lr10*), Krasnozerka (with *Lr1* and *Lr3*), Nerda, Orenburgskaya yubileynaya (with *Lr10*), Orenburgskaya 23 (with *Lr1* and *Lr3*), Salavat Yulayev (with *Lr1*, *Lr3*, and *Lr34*), Uralosibirskaya, Uralosibirskaya 2, Zaural'skaya zhemchuzhina (with *Lr1*, *Lr3*, and *Lr10*), and Zaural'skiy yantar (with *Lr3* and *Lr10*). Cultivars with *Lr19* (including Ekada 113, Kinel'skaya Niva, L503, L505, and Tulaikovskaya 108) are also grown in the region. In 2020, the new cultivar Zaural'skaya volna, with *Lr19* combined with the lowly effective genes, *Lr3* and *Lr10*, was released for production in the Urals and Western Siberia.

The widespread production of genetically protected cultivars in the Ural region led to significant changes in the pathogen population observed in 2010 in comparison to previous years. A high-virulence polymorphism of *P. triticina* was recorded in genotypes with *Lr9*,

*Lr26*, and *Lr19*. A decrease in prevalence virulence to *Lr26* and an increase to *Lr9* was also recorded. Prevalence of virulence to *Lr19* was high before 2010 but has subsequently become rare or undetectable. A single *Lr24*-virulent isolate of *P. triticina* was found in 2008 but no others have been found since. The frequencies of virulence to other lines used in the analysis (Tc: *Lr1*, *Lr2a*, *Lr2b*, *Lr2c*, *Lr3a*, *Lr3bg*, *Lr3ka*, *Lr10*, *Lr14a*, *Lr14b*, *Lr15*, *Lr17*, *Lr18*, *Lr20*, and *Lr30*) were persistently high (75–100%)

**Western Siberia.** Over 30% of the spring wheat cultivars in the State Register are recommended for the Western Siberian region. In 1995, cultivar Tertsia with the *Lr9* gene was the first resistant cultivar included in the Register. Initially, this cultivar was thought to carry a new *LrTr* gene [8]; however, in the mid-2000s, molecular testing showed that the *LrTr* gene is identical to the *Lr9* gene [61]. In the 2000s, the number of cultivars with the *Lr9* gene recommended for use in Western Siberia increased significantly (including Altayskaya 110 with *Lr3* and *Lr10*; Apasovka with *Lr10*; Sonata and Udacha combined with *Lr3*; Aleksandrina and Novosibirskaya 44 with *Lr1* and *Lr10*; Sibakovskaya yubileynaya with *Lr1*, *Lr3* and *Lr10*; Mariya 1 with *Lr3* and *Lr10*; Sibirskiy al'yans with *Lr1*; Novosibirskaya 18 with *Lr10*; Novosibirskaya 44 with *Lr1*, *Lr3*, and *Lr10*; Sibirskaya 17 and Stolypinskaya with *Lr10*; and Start with *Lr34*). Tc*Lr9* virulence in *P. triticina* was first recorded in Western Siberia in 2007 [62]. In subsequent years, the prevalence of virulence steadily increased. In 2010, cvs Omskaya 37 and Omskaya 38, with the effective combination of *Lr19* and *Lr26*, were released for production in the region. However, both *Lr19* and *Lr26* have since lost their effectiveness in the region. Virulence in *P. triticina* to *Lr26* in Western Siberia has increased over the years of research, from 20 to 100%, and to *Lr19* from 0 to 14% [58]. However, no *P. triticina* with virulence to both of these genes has been isolated.

In 2020, the Russian wheat cultivar Lider 80 with *Lr24* was released for the first time for production in the Western Siberian and Eastern Siberian regions [63] and the gene has proven to be effective against leaf rust in Western Siberia [64]. Only the single isolate of *P. triticina*-virulent *Lr24* was obtained in Western Siberia in 2007 and 2009.

Cultivars grown in Western Siberia, as elsewhere in Russia, commonly have *Lr1* (20%), *Lr3* (35%), *Lr10* (50%), and *Lr26* (10%) genes, and the partial resistance gene *Lr34* (8%). Many cultivars have combinations of genes with low effectiveness, including Altayskaya 325, Bel', Kantegirskaya 89, Melodiya, Omskaya 33, Polyushko, Pamyati Vavenkova and Tulunskaya 12, combined with *Lr1*, *Lr3*, and *Lr10*; Boyevchanka with *Lr3*, *Lr10*, and *Lr26*; Omskaya 29, Omskaya yubileynaya, Svetlanka, and Stepnaya volna with *Lr3*, *Lr10*, and *Lr34*; Element 22, OMGAU 100 with *Lr26* and *Lr10*; Stolypinskaya with *Lr3* and *Lr34*; Altayskaya 70, Altayskaya 105, Ekstra, Novosibirskaya 29, Novosibirskaya 22; Obskaya 2 and Pamyati Afrodity with *Lr3* and *Lr10*; Omskaya 42, Gornoural'skaya and Katyusha with *Lr10* and *Lr34*; Altayskaya 99 and Tobol'skaya with *Lr1* and *Lr10*; and Altayskaya zhnitsa with *Lr1* and *Lr3*.

Many Western Siberian cultivars also have *Lr34*, transferred from Strela (*Lr34* and *Lr10*) and Skala (*Lr34* and *Lr3*). These cultivars were commonly grown in the 1970s to 1990s [65]. Cultivar Strela is one of the best Russian cultivars and, in some years, has been produced on over 1 million ha. Strela and Skala have been actively used in breeding, which has led to the widespread use of *Lr34* in current spring wheat cultivars.

The main change in the Western Siberian populations of *P. triticina* in 2010, as in Ural populations, was the increased prevalence of virulence to *Lr9*. Since all *Lr9*-virulent isolates were avirulent to *Lr26*, there was a concurrent decrease in virulence to *Lr26*. The Western Siberian populations of *P. triticina* were characterized by a higher number of virulence alleles than the European populations. Virulence to *Lr1* and to most of the less effective genes was consistently high in all years of the research.

## 3. Conclusions

Research on wheat cultivars and leaf rust pathogen populations in Russia in 2000–2020 demonstrated the continuous evolution of *P. triticina* in response to wheat breeding efforts.

Populations of the pathogen were characterized by higher variation in virulence in regions where cultivars with oligogenic resistance are commonly grown.

Here, we have documented significant progress in the development of leaf rust-resistant cultivars for Russia. The identification of resistance genes has allowed a greater understanding of the virulence dynamics in *P. triticina* populations and helped to analyze the host–pathogen interactions. Juvenile genes (*Lr24*, *Lr28*, *Lr29*, *Lr41*, *Lr42*, *Lr45*, *Lr47*, *Lr50*, *Lr51*, *Lr53*, and *Lr57*) show high efficiency in Russia and can serve as a basis for the breeding of rust-resistant cultivars. Some of these genes (*Lr24*, *Lr41*, and *Lr47*) are already used by Russian breeding centers. Furthermore, donors of new foreign genes from *Th. intermedium*, *Ae. speltoides*, *Ae. tauschii*, *T. durum*, and others are widely used in breeding. Effective genes used in combination with genes that are no longer adequately effective on their own makes it possible to extend their useful lifetime (e.g., *Lr9*, *Lr19*, and *Lr26*).

In the middle of the last century, cultivars such as Rieti, Kichener, Kanred and Klein 33, Neuzucht, Selkirk, Lee, Timshtein, RedRiver 68, Norman, Sonora 64, Lerma Rojo, Mentana, Maria Escobar, and Supremo 211 were commonly used for breeding in Russia. They provided the widespread use of *Lr1*, *Lr2*, *Lr3*, *Lr10*, *Lr14b*, *Lr17*, *Lr16*, and *Lr23* in Russian cultivars [66]. Using molecular markers, the wide distribution of *Lr1*, *Lr3*, and *Lr10* genes in Russian cultivars was confirmed. The lack of reliable diagnostic markers for other *Lr* genes makes it impossible to assess their occurrence in Russian cultivars.

Data on pathogen population studies and the occurrence of *Lr* genes in wheat cultivars should be considered by regional breeding programs and for the release of new, genetically protected cultivars. The development of new genotypes, along with donors of effective *Lr* genes, needs to use effective combinations of genes (pyramiding). Molecular markers selected for use in marker-assisted selection have greatly facilitated a successful response to this challenge.

**Author Contributions:** Conceptualization, E.G.; methodology, E.G. and E.S.; software, E.G. and P.G.; validation, E.G. and E.S.; formal analysis, E.G. and P.G.; investigation, E.G., E.S.; data curation, E.G.; writing—original draft preparation, E.G. All authors have read and agreed to the published version of the manuscript.

**Funding:** This study was funded by the Russian Science Foundation (project number 19-76-30005).

**Institutional Review Board Statement:** Not applicable.

**Informed Consent Statement:** Not applicable.

**Data Availability Statement:** All data are provided in the manuscript.

**Acknowledgments:** Editorial support of Ian Riley is highly appreciated.

**Conflicts of Interest:** The authors declare no conflict of interest.

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
