# Peer review of "Leaf Rust Resistance Genes in Wheat Cultivars Registered in Russia and Their Influence on Adaptation Processes in Pathogen Populations"

_agriculture, doi:10.3390/agriculture11040319_

Round 1

Reviewer 1 Report

The general information and history were well introduced in manuscript.

There are some comments.

1.  First paragraph in page 1. 
- The author well explained the situation in Russia and the damage to leaf rust. It would be good to add by comparing the current status of other neighboring countries.

2. " Russia’s large size means there is a wide diversity of soil types and agricultrural environments" in page 2.
- Then, it would be good if additional explanations were added about the characteristics of typical Russian agricultural land soil (soil classification).

3. " Over the last decade in Russia, standard breeding methods have been enhanced by the adoption marker-assisted breeding, and this innovation has delivered great efficiencies for the development of new cultivars. "
- What is the standard breeding method? More specifically information be added.

4.   [15-16, 20] in page 3.
- Correct to [15,16,20]

5. Main section in review article.
- It seems to have been well explained by dividing the characteristics by region.

6. Table 2.
-In rows 3 and 4, the year 2010 overlaps. It will have to be revised to 2009 or 2011.

7. " This study was funded by the Russian Science Foundation (project number 19-76-30005). Editorial support of Dr. Ian Riley is highly appreciated. "
-This sentence needs additional bold "Funding" title or "Acknowledgement" title.

8. Summary
How about changing "Summary" to "Conclusion"?
And delete the first paragraph?

Author Response

Dear reviewer,

We accepted all editorial-type changes. A revised version of our manuscript was prepared accordingly.

Thank you very much. With kind regards,

Elena Gultyaeva

Answers to comments.

  1. First paragraph in page 1. 
    - The author well explained the situation in Russia and the damage to leaf rust. It would be good to add by comparing the current status of other neighboring countries.

It was added. A decrease in the importance of  P. triticina compared to P. graminis and P. striiformis in 2000 was also noted in the other countries, including neighboring to Russia (Kazakhstan, Armenia, Azerbaijan, Georgia, Kazakhstan, Kyrgyzstan, Turkmenistan, Uzbekistan etc.) [4, 5]. 

  1. " Russia’s large size means there is a wide diversity of soil types and agricultrural environments" in page 2.
    - Then, it would be good if additional explanations were added about the characteristics of typical Russian agricultural land soil (soil classification).

We deleted the word “soil” from the sentence. Because it is not so important information for such studies.

  1. " Over the last decade in Russia, standard breeding methods have been enhanced by the adoption marker-assisted breeding, and this innovation has delivered great efficiencies for the development of new cultivars. "
    - What is the standard breeding method? More specifically information be added.

It was changed “Over the last decade in Russia, breeding have been enhanced by the adoption marker-assisted breeding, and this innovation has delivered great efficiencies for the development of new cultivars"

  1.   [15-16, 20] in page Correct to [15,16,20] and 6. Table 2. -In rows 3 and 4, the year 2010 overlaps. It will have to be revised to 2009 or 2011.

Corrected.

  1. " This study was funded by the Russian Science Foundation (project number 19-76-30005). Editorial support of Dr. Ian Riley is highly appreciated. " -This sentence needs additional bold "Funding" title or "Acknowledgement" title.

Acknowledgement was added.

  1. How about changing "Summary" to "Conclusion"?
    And delete the first paragraph?

We replaced "Summary" to "Conclusion" and  changed the first and second sentences.

Reviewer 2 Report

Wheat leaf rust, caused by Puccinia triticina, is a fungal leaf disease that can pose a significant threat to the yield and quality of wheat crops worldwide. This review focused on the genetic diversity of winter and spring wheat cultivars and their impact on the prevalence of virulence in pathogen populations in different regions of Russian and provided a good overview of the past and current status of leaf rust in wheat production. I advise several minor revisions.

Line 13 us should be use

Line 17 phenotyping

Line 25 Lr, Puccinia triticina, Triticum aestivum should be in italic type

Line 42 and 44, fourfold and two and half times?  How did you calculate the numbers?

Line 93 was should be there was

Line 127 s should be deleted

Line 134 stern rust should be leaf rust

Table 1 The quality of this table must be improved. The marker type and the distance to the target gene should be included.

Line 157 r34 should be Lr34

Line 169 and should be any

Line 370 demonstrate should be demonstrated

Line 371 P. triticina should be of P. triticina

Author Response

Dear reviewer,

We accepted all editorial-type changes and added data to table.

A revised version of our manuscript was prepared accordingly.

Thank you very much.

With kind regards,

Elena Gultyaeva 

Line 42 and 44, fourfold and two and half times?  How did you calculate the numbers?

Corrected: The cultivar’s number was taken from the official information published in State Register of Breeding Achievements and recommended for cultivation in Russia 1996, 2020. Sentence was corrected “number of winter wheat cultivars included in the State Register of Breeding Achievements and recommended for cultivation in Russia in 2020 increased threefold compared to the mid 1990s (333 in 2020 vs 113 in 1996), and spring wheat cultivars by twofold (261 vs 138). It was fourfold changed to threefold and two and half times to twofold.

Round 2

Reviewer 1 Report

Review comments are well revised and reflected.